# Review of the New Zealand *Theileria orientalis* Ikeda Type Epidemic and Epidemiological Research since 2012

**DOI:** 10.3390/pathogens10101346

**Published:** 2021-10-19

**Authors:** Kevin Lawrence, Kristene Gedye, Andrew McFadden, David Pulford, Allen Heath, William Pomroy

**Affiliations:** 1School of Veterinary Science, Massey University, Palmerston North 4100, New Zealand; K.Gedye@massey.ac.nz (K.G.); W.Pomroy@massey.ac.nz (W.P.); 2Ministry for Primary Industries, P.O. Box 40742, Upper Hutt 5018, New Zealand; andrew.mcfadden@mpi.govt.nz (A.M.); david.pulford@mpi.govt.nz (D.P.); 3Hopkirk Research Institute, AgResearch Ltd., Private Bag 11008, Palmerston North 4442, New Zealand; acgandmm.heath@actrix.co.nz

**Keywords:** *Theileria orientalis*, Ikeda, New Zealand, *Theileria* associated bovine anaemia, bovine

## Abstract

This article sets out to document and summarise the New Zealand epidemic and the epidemiological research conducted on the epizootic of bovine anaemia associated with *Theileria orientalis* Ikeda type infection, which began in New Zealand in August 2012. As New Zealand has no other pathogenic tick-borne cattle haemoparasites, the effects of the *T. orientalis* Ikeda type infection observed in affected herds and individual animals were not confounded by other concurrent haemoparasite infections, as was possibly the case in other countries. This has resulted in an unbiased perspective of a new disease. In addition, as both New Zealand’s beef and dairy cattle systems are seasonally based, this has led to a different epidemiological presentation than that reported by almost all other affected countries. Having verified the establishment of a new disease and identified the associated pathogen, the remaining key requirements of an epidemiological investigation, for a disease affecting production animals, are to describe how the disease spreads, describe the likely impacts of that disease at the individual and herd level and explore methods of disease control or mitigation.

## 1. The New Zealand Epidemic

The New Zealand *Theileria orientalis* Ikeda type epidemic began in 2012 with index cases identified during the spring calving period on two dairy farms, in Kamo, Northland, on the 22 August 2012 and in Reporoa, Waikato, on the 30 August 2012 [1]. 

Although initially identified on two dairy farms, the epidemic was at the beginning largely restricted to mostly beef farms in Northland but by the autumn of 2013 had spread to autumn calving dairy herds in the Waikato, probably when infected pregnant dairy cows returned from summer grazing in Northland. By the following spring of 2013 the outbreak had spread over most of the upper North Island [2], requiring the development of rapid quantitative methods of diagnosis [3]. *Theileria* associated bovine anaemia (TABA) remained notifiable in New Zealand until 30 November 2014, during which time the Ministry for Primary Industries (MPI) collected an almost unique data set, which has resourced much of the New Zealand research conducted on *T. orientalis* Ikeda type since then. 

The epidemic curve constructed using these data (Figure 1), showed a pattern of seasonal peaks occurring in the spring and autumn; with the spring peak being the greatest [4]. Descriptive data collected by MPI allowed the epidemic curve to be categorised by age, (calf less than 6 months of age or adult), and by farm type, (dry stock, dairy, or beef). Categorisation showed that dairy and adult were the most common classes of cattle affected in the epidemic (Figure 1).

The rapid spread through the upper North Island during 2013 [2], was aided by the unique New Zealand policy whereby share milkers change contracts and dairy farms on the 1 June each year, colloquially known as “gypsy day” or “moving day” A share farmer is a contractor who manages a dairy farm with or without owning the animals, based on a share of the milk income. On “moving day”, each year, thousands of share farmers, their families, and their dairy herds move farms, often over several hundred kilometres. In June 2013 this had the effect of translocating naïve herds into areas where infected ticks were already present and moving infected herds into areas where ticks were still naïve. As a result, there was a rapid acceleration in the spread of disease throughout the upper North Island during spring 2013 (Figure 2). Confirmation of this rapid spread was provided by the PCR testing of stored bloods from pre-export sampling of dairy heifers, and bloods taken as part of a voluntary bovine viral diarrhoea (BVD) control programme. These bloods were collected between November 2012 and June 2013. The results clearly showed that in less than 12 months from the index cases, the disease had spread to many regions of New Zealand, with the highest prevalence of infected herds being in Northland (94%), and Auckland and Waikato (33%) [5]. A small number of Ikeda positive herds was even found in regions of the South Island where ticks are absent, likely as the result of the movement of cattle previously exposed to infection in the North Island. “Moving day” 2014 achieved a further rapid acceleration of case farms (Figure 2). However, during the whole period that MPI records were kept (22 August 2012 to 20 November 2014) only one PCR positive farm was identified in the South Island, and this was not until late 2014 [4]. Following this initial case, TABA then became more common and was regularly diagnosed in the Golden Bay area at the northern tip of the South Island by 2018 [6].

It became very evident, early in the epidemic, that many farms were becoming infected without farmers reporting clinical disease outbreaks. A prevalence study conducted by MPI in 2013 supported this assumption showing that on average 87% of cows were already infected *T. orientalis* Ikeda when the first clinical cases were reported in that herd [7]. At present in 2021 *T. orientalis* Ikeda remains endemic in all these original areas with veterinarians still reporting clinical cases but not the mass numbers seen with the original epidemic.

## 2. Source of Infection

It is likely that the live importation of infected cattle from Australia, prior to 2012, was responsible for the introduction of the Ikeda type into New Zealand. A previous outbreak of *T. orientalis* in the 1982–1986 was also attributed to live imports from Australia [8]. Retrospective testing of bloods collected annually from sentinel farms in Northland and the Waikato since 2008, as part of a MPI arbovirus surveillance program, showed that 1/7 farms were *T. orientalis* Ikeda PCR positive in December 2011 and by December 2012, 6/7 farms were PCR positive (Figure 3). This indicates that *T. orientalis* Ikeda infected cattle arrived in Northland prior to December 2011. Until August 2017 New Zealand did allow live cattle imports from Australia and records reveal that approximately 90 live cattle were imported into New Zealand from Australia in the 10 years prior to 2012 (MPI data on file). Which particular importation was responsible is not clear and the possibility remains that multiple disease incursions may have occurred.

## 3. Tick Distribution

*Haemaphysalis longicornis* is the only known competent vector for *T. orientalis* in New Zealand [9]. This tick species was introduced into New Zealand possibly in the late 19th or early 20th century [9] and is now considered to have a stable distribution which is widespread in the North Island, occurring as far south as Waikanae in the west and the Wairarapa in the east, it is also established in the top of the South Island [9]. In New Zealand the *H. longicornis* life cycle is usually completed within 12 months, with over-wintering nymphs mainly engorging from July to September, adults from November to December and larvae from February to April [9]. Suitable hosts include cattle, deer, goats, horses, hares, sheep, dogs, kiwis, and even humans [9,10]. Figure 4 represents the most recent distribution map for *H. longicornis* in New Zealand and was based on the results of a comprehensive telephone survey, field observation and tick surveillance on deer [4,9].

## 4. Spread of Disease

The main method of disease dissemination in the 2012 New Zealand TABA epidemic was the movement of parasitaemic cattle into naïve herds, and the consequences of those movements on the epidemiology of the disease are summarized in Figure 5. Infected farms were 2 times more likely to have had cattle movements onto their farms than uninfected farms [2] and the ramifications of that movement depended on the tick density of the area in which the naïve farm was situated. In high tick density areas, endemic stability rapidly established with the disease more often seen in young cattle rather than older cattle. In Northland, which was probably the first region in New Zealand to achieve endemic stability, beef or dairy calves were 26 times more likely to be diseased than calves from elsewhere in the North Island [2].

Local spread of infection to neighbouring farms from recently infected herds is also highly likely in high tick density areas. The epidemic curve described a propagating epidemic in the early stages in Northland and strong temporal-spatial clustering of case farms was also seen [2]. New infections were reported on farms 20–30 days after infection was reported on a nearby farm, situated 5–20 km away [2]. Both findings support the importance of local spread in the early stages of *Theileria* epidemic. 

The severity of disease and the age group affected, also depends on the tick density. The environmental suitability of tick habitats falls as the locations move further south in the North Island [11]. Consequently, the further south that infection occurs then the more likely disease will be seen in adult cattle rather than calves [12] often with a more severe clinical disease outcome. Figure 6 shows that as latitude south increases, the proportion of TABA cases in calves decreases to almost zero in the lower North Island. In areas outside of known tick areas, where ticks are absent, there is still a low probability of non-tick associated disease transmission [13]. However, with no reports of TABA in nine years from these tick-free areas it is unlikely that acute clinical disease occurs after mechanical or iatrogenic spread. It is also unlikely that the distribution of tick habitats in New Zealand will be greatly affected by global warming, although some increase in TABA disease frequency may be expected on the West Coast of the South Island [11]. Trans-placental infection is not shown in Figure 5 as trans-placental infection occurs at a very low level, if at all, in infected cattle and has a very minor role in the epidemiology of TABA [14].

## 5. Impact of Disease

Before the emergence of *T. orientalis* Ikeda type in 2012 it was believed that New Zealand cattle were only parasitised by the more benign *T. orientalis* types, Chitose and Buffeli, with just sporadic disease outbreaks reported [8,15,16]. Despite the higher pathogenicity of *T. orientalis* Ikeda type [17,18] most New Zealand farms affected in the 2012 Ikeda epidemic recorded relatively low mortality and morbidity rates, the medians being 0.23% and 0.97%, respectively [19]. These disease estimates were based on survey data from 196 dairy and beef farms and remain the only published estimates of morbidity and mortality rates for *T. orientalis* Ikeda type outside of Japan.

Infection with *T. orientalis* Ikeda is lifelong and peak infection levels are usually reached at 4 to 6 weeks after infected ticks have fed (Figure 7). Following the peak, the infection levels drop considerably, and the animal enters an asymptomatic carrier state, probably for the rest of its life. In the following text, peak infection will be referred to as the “acute infection or acute phase” and the carrier stage will be referred to as the “chronic infection or chronic phase”.

In the early part of the 2012 epidemic, diagnosis by veterinarians in the field was based on clinical signs, measuring the packed cell volume (PCV) and submitting blood smears to the private regional veterinary laboratories for Giemsa staining. Later, Giemsa staining was mostly replaced by PCR testing. The most common clinical signs recorded for cattle with acute infection in New Zealand are jaundice, lethargy, pale mucous membranes, anaemia, and reduced milk production [21]. In the chronic phase, infection is asymptomatic except for the hypothesised syndrome of ill thrift, diarrhoea, and death reported in weaned beef calves [21]. In the earlier 1982–1986 *T. orientalis* epidemic the two most common presenting signs from 101 cases, were ill-thrift (29/101) and diarrhoea (23/101) [22]; it is noteworthy that anaemia and jaundice were not commonly reported in that epidemic [22]. Although never proven conclusively, the 1982–1986 New Zealand epidemic was likely associated with the Chitose type and the agreement between two different *T. orientalis* types, Ikeda and Chitose, may lend some support to the supposition that the pathological effects of *T. orientalis* infection are not just limited to extravascular haemolytic anaemia, and in the absence of anaemia, the Ikeda and Chitose types can potentially both produce similar disease syndromes of ill-thrift and diarrhoea. The actual prevalence of the Chitose type prior to the Ikeda epidemic is unknown as no specific New Zealand studies or population surveys were undertaken.

The most common haematological and biochemical changes seen in the acute phase of infection were similar in both adult cattle and young calves [23]. These were noted as a regenerative extravascular haemolytic macrocytic anaemia, resulting in reduced haematocrit and RBC counts, and associated biochemistry changes of elevated GGT and bilirubin levels [23]. However, calves were less likely to develop severe anaemia and adult cattle were more likely to show anoxic liver damage at low HCT, with increased levels of AST and GLDH and an inflammatory response [23]. At the individual cow level, anaemic animals took approximately 53 days to recover [24] and at the herd level it took about 80 days for the herd prevalence of anaemia to return to zero, following diagnosis [7].

The potential outcomes of *Theileria orientalis* Ikeda infection in naïve animals are shown in Figure 8. The concepts conveyed in Figure 8 are that *T. orientalis* Ikeda type infection is a necessary but not sufficient cause of disease (anaemia, reduced milk production and reduced liveweight gain) and that other contributory causes, such as stress (physiological and nutritional), and concurrent disease (e.g., BVD), have a strong influence on the outcome of infection, at both the individual and the herd level. Sex was also found to influence the infection intensity with female beef calves developing a higher infection than male calves [20]. Breed and age are also likely to impact the infection intensity and thus the clinical outcome, purebred Friesians showed a higher prevalence of clinical theileriosis than crossbred dairy cattle (predominantly Jersey crossed with Friesian in different proportions) [25].

Very few chronically infected cows appeared to recrudesce into acute illness despite nutritional stress or severe concurrent disease [26]. To emphasise this, broken grey lines and arrows were used in Figure 8 to show a weak association in the direction of the arrow. Instead, it is likely that the role of *T. orientalis* Ikeda type in disease presentation changes from the acute to the chronic phase of infection. In the acute phase the effects of the contributory causes influence the pathophysiological response to *T. orientalis* Ikeda infection leading to higher infection intensities and more severe disease outcomes, i.e., the contributory causes help set the conditions under which we see acute TABA. Whereas in the chronic phase *T. orientalis* Ikeda type infection now interacts with these contributory causes, leading to potentially more severe effects resulting from stress (physiological or nutritional) or concurrent disease, on milk production and liveweight gain than would be seen in uninfected cattle, i.e., the chronic *T. orientalis* Ikeda infection decreases the fitness of the animal to tolerate other endemic diseases and deficiencies. This reduction in fitness brought on by chronic infection is the likely explanation for the hypothesised syndrome of ill thrift, chronic diarrhoea and death reported in weaned beef calves [21]. These calves present with quite vague illnesses which are likely related to parasitism, concurrent infections such as BVD or *Yersinia pseudotuberculosis* and poor nutrition exacerbated by the chronic *Theileria* infection.

It is likely that adverse production effects result from the combined effects of both anaemia and anorexia. The Northland index case had approximately 20 cows die during the calving season, had a six-week in-calf rate of only 44% (target 78%) during the mating season and the per-cow milk season production was just 230 kg milk solids, well down on the regional average of 315 kg [27]. In another outbreak where naïve cattle were moved to an endemic area, costs of more than $NZ 1 million were incurred by a single herd [28]. However, on many farms the effects were less dramatic. Nevertheless, modelling carried out by veterinarians at MPI in May 2013 showed that the average farm impact of *T. orientalis* Ikeda infection was estimated at $14,000, but the range was $4000–$29,000 [27]. The full cost of the 2012 epidemic in New Zealand is unlikely to ever be accurately estimated, since herds were often almost 100% infected, over a very short period [7]. This meant that uninfected controls for matching to and contrasting with infected animals were seldom available for herd-based production studies. The loss in milk production, for non-clinical incidence cases, has been estimated at around 20 kg milk solids per cow [26], and for live weight gain at 2 kg/week for 2 years old Friesian bulls [29] and at 900 g/week for beef calves [20]. A negative effect of acute infection on reproduction was also found, infected cows had a higher empty (non-pregnant) rate and a lower conception rate than uninfected cows, with losses estimated at $196/cow [25]. There was no effect of infection on male fertility observed, with semen quality in ten 2-year-old Friesian bulls, experimentally infected with *T. orientalis* Ikeda, remaining unchanged [30]. However, during the acute phase of infection, when the infection intensity was rapidly increasing, the infected bulls took a longer time to repeatedly mount females and were less dominant in the herd social hierarchy compared to uninfected controls [30].

## 6. Modelling to Describe the Distribution of the Tick *Haemaphysalis longicornis* and the Likely Extent of the TABA Epidemic

Two models were constructed to explain the spatial distribution of ticks and disease transmission in New Zealand. The first model was a simple climate model to determine the spatial extent of habitat suitability for *H. longicornis* in New Zealand [11]. The model was developed using a rule-based climate envelope model based on the environmental requirements for off-host tick survival [31,32]. The model predicted that 75% of cattle farms in the North Island, 3% of cattle farms in the South Island and 54% of cattle farms in New Zealand overall had habitats potentially suitable for the establishment of *H. longicornis* (Figure 9a).

The second model used the Maxent (maximum entropy) modelling program [33,34] to predict the relative environmental suitability for *T. orientalis* transmission throughout New Zealand [13]. The Maxent model predicted that 99% of North Island cattle farms, 64% South Island cattle farms and 89% of New Zealand cattle farms overall could potentially be suitable for *T. orientalis* transmission (Figure 9b). The areas of Figure 9b identified as having high suitability for disease transmission are likely to achieve endemic stability, whereas in the low suitability areas, endemic stability will not be achieved, and disease will continue to be sporadic and potentially severe when it does occur. 

## 7. Managing the New Zealand Epidemic

Several resources were developed for veterinarians and farmers to manage outbreaks by different national bodies collaborating, including MPI together with the New Zealand Veterinary Association and DairyNZ. These resources included the FANI card [35], the *Theileria* veterinary handbook [36] and the *Theileria* veterinary handbook 2 [28].

The FANI (Field Anaemia Nearest Indicator) card (Figure 10 and Figure 11), was developed by MPI in 2013, in which colour charts were used to assess the level of anaemia in individual animals, based on examining the mucous membrane of the vulva [35]. The FANI card has four colour categories, which span the range from healthy to severely anaemic animals, and a fifth category differentiating anaemia due to blood loss. This system was designed to provide enough representation of the full range of disease signs but be simple enough to facilitate meaningful clinical classification. It enabled a rapid cow-side interpretation and treatment decision to be made for each individual cow [35]. Cattle identified as having a PCV < 10% to 14% using the FANI card, often had a confirmatory PCV carried out by veterinary clinicians or regional laboratories. Those cattle with a PCV below 0.12 would likely have died without a blood transfusion and during the New Zealand TABA epidemic thousands of blood transfusions were given by veterinarians [35,37]. Anecdotal reports indicate that dairy practitioners are still required to transfuse severely ill cattle from time to time but not in the same numbers as during the epidemic.

The *Theileria* veterinary handbook [36] and the *Theileria* veterinary handbook 2 [28] collected material that had previously been printed in VetScript [35,38,39,40,41] and was designed as a ‘ready-to-use’ reference for veterinarians and the agricultural sector, which provided up to date information at the time of publication. 

## 8. Control of Disease

Despite widespread promotion and uptake, the use of pour-on acaricides proved largely ineffective at preventing oriental theileriosis in the 2012 epidemic [42]. This was not surprising given that on average 87% of cows were already infected when the first cases of TABA are reported in that herd [7]. Currently those New Zealand cattle farms in the low-tick density areas which fail to reach or maintain endemic stability are the farms that continue to experience acute clinical theileriosis in older cattle [26]. The most important control method that can be implemented on these farms is the deliberate exposure of young cattle to infection, to force the herd into endemic stability. Executing this program should not result in increased clinical cases of TABA, since in the acute phase young cattle appear to be less severely affected than older cattle [23], although some effect on growth rate of young stock might be expected [20]. However, once the young cattle reach the chronic infection phase, recrudescence into acute disease is rarely seen [26].

There is still an ongoing requirement in New Zealand for an effective chemotherapeutic agent to treat oriental theileriosis, with reports of continued high mortality rates in beef calves in Northland (L. Bawden, pers comm.). Despite favourable anecdotal reports, toltrazuril proved ineffective in a clinical trial [43] and although buparvaquone has shown some reported efficacy in the field [44] the results are often variable. In one New Zealand trial, although with small numbers of treated animals (*n* = 45), there was an 80% reduction in case fatality rates and a 40% reduction in milk loss in treated cattle compared to untreated controls [26]. However, the long legal withhold periods for meat (18-month) and milk (43-day) currently enforced in New Zealand for this drug, severely restrict its usage on most farms [45].

## 9. Future Research

As endemic stability is reached in many regions the incentive for further research has dropped, however there are two pieces of work that would complement that already completed. The first would be to investigate whether latent carriers, such as sheep, deer, dogs, or horses, played a significant role in the rapid spread of TABA through the North Island, by providing additional sources of infection for naïve ticks. Although this knowledge would have little bearing on the current New Zealand epidemic it would help explain how the epidemic spread so quickly and may aid disease control in new outbreaks, possibly overseas. A recent study has examined the role of sheep in the spread of *T. orientalis* [46]. In this study it was found that some sampled sheep in endemic areas had low levels of *T. orientalis* Ikeda type DNA and that naïve tick larvae which fed on these sheep could become infected, possibly through co-feeding with other tick stages. Further work will need to explore whether these infected ticks can spread active infection to naïve cattle and close the loop, before the impact of sheep on the epidemiology of theileriosis can be assessed. The second would be to identify alternate chemotherapeutic agents to buparvaquone, ones with a shorter, more reasonable, withholding period. There is still a significant burden of clinical disease in the unstable endemic areas of New Zealand and in naïve heavily pregnant dairy cows moving from disease free areas to infected areas, as has been seen post “moving day”. An effective chemotherapeutic agent would dramatically reduce animal suffering and lost production if it were available.

Eradication of the Ikeda type would be extremely difficult due to the widespread distribution, involvement of other ruminants as peripheral hosts and the absence of an effective treatment. The only likely option would be test and slaughter and that is clearly untenable given the number of animals involved.

Until now there has been no interest in developing a vaccine against *T. orientalis* Ikeda type in New Zealand, however, this may change since a recent Australian study found that prior infection with *T. orientalis* Buffeli may be protective [47]. If this is the case, then it is possible that the widespread infection of cattle in the upper North Island with the Buffeli and Chitose types, that was presumed to exist prior to the 2012, may have resulted in a less severe epidemic than other countries have experienced. 

Important lessons that other countries can learn from the New Zealand epidemic are that if you have a competent vector such as *H. longicornis*, and if you import infected cattle, the disease will quickly become established and spread rapidly. The current situation in the USA is slightly different in that they had small, isolated pockets of infected cattle but no competent vector, so no disease was seen until the introduction of *H. longicornis* in 2017 [48,49].

## Figures and Tables

**Figure 1 pathogens-10-01346-f001:**
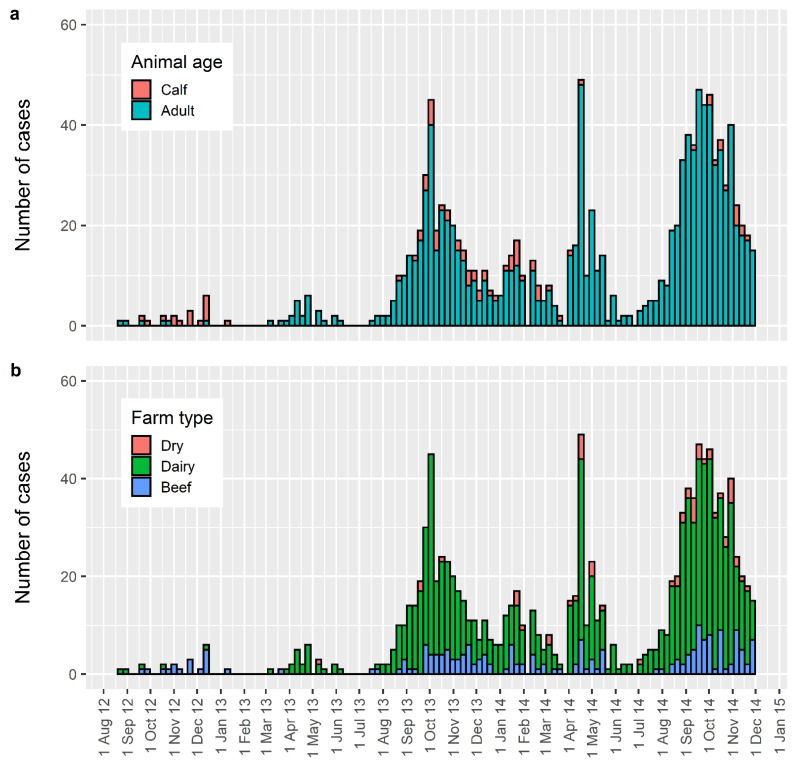
Epidemic curve for confirmed cases (“Ikeda PCR positive”) for the period 1 August 2012–1 December 2014. (**a**) Shows the epidemic curve categorised by age, calf (less than 6 months of age) or adult. (**b**) Shows the epidemic curve categorised by farm type, dry stock, dairy, or beef.

**Figure 2 pathogens-10-01346-f002:**
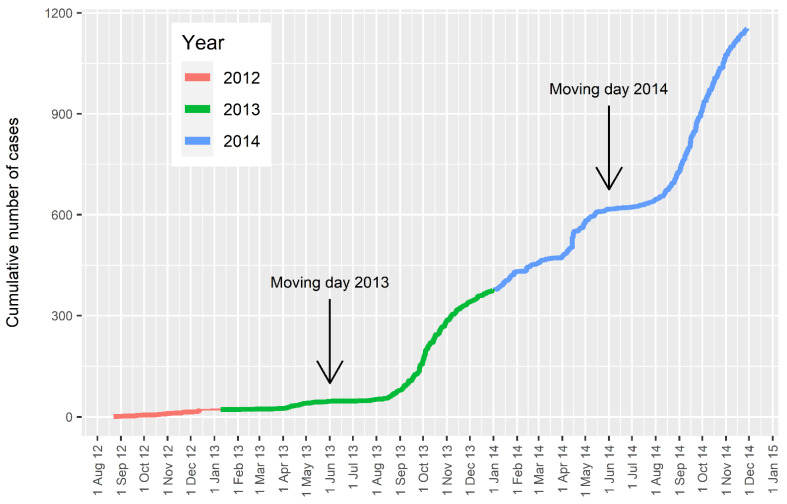
The cumulative number of cases 1 August 2012–1 December 2014 categorised by year, showing the moving day for 2013 and 2014.

**Figure 3 pathogens-10-01346-f003:**
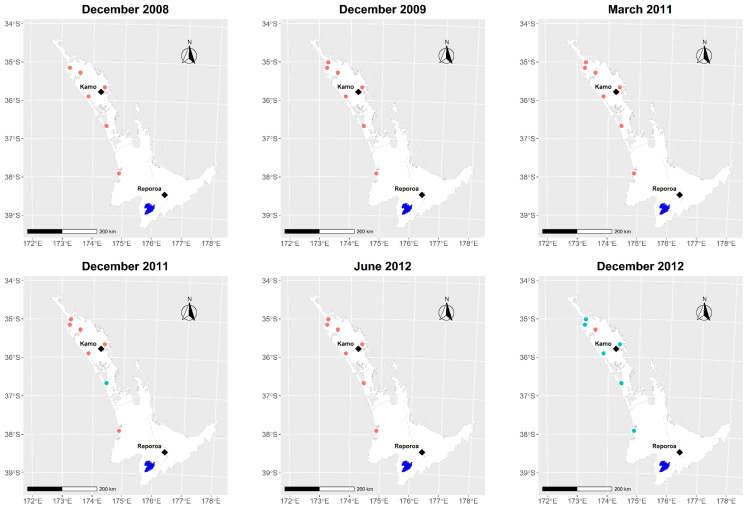
Results from PCR testing of stored bloods from seven sentinel herds in Northland and Waikato, sampled as part of an arbovirus surveillance program, sampled from 2008 to 2012. Red dots show PCR negative farms and green dots show PCR positive farms for *Theileria orientalis* Ikeda. The two-index case dairy farms in Reporoa and Kamo are shown (◆). (Only 6 herds were tested in December 2008).

**Figure 4 pathogens-10-01346-f004:**
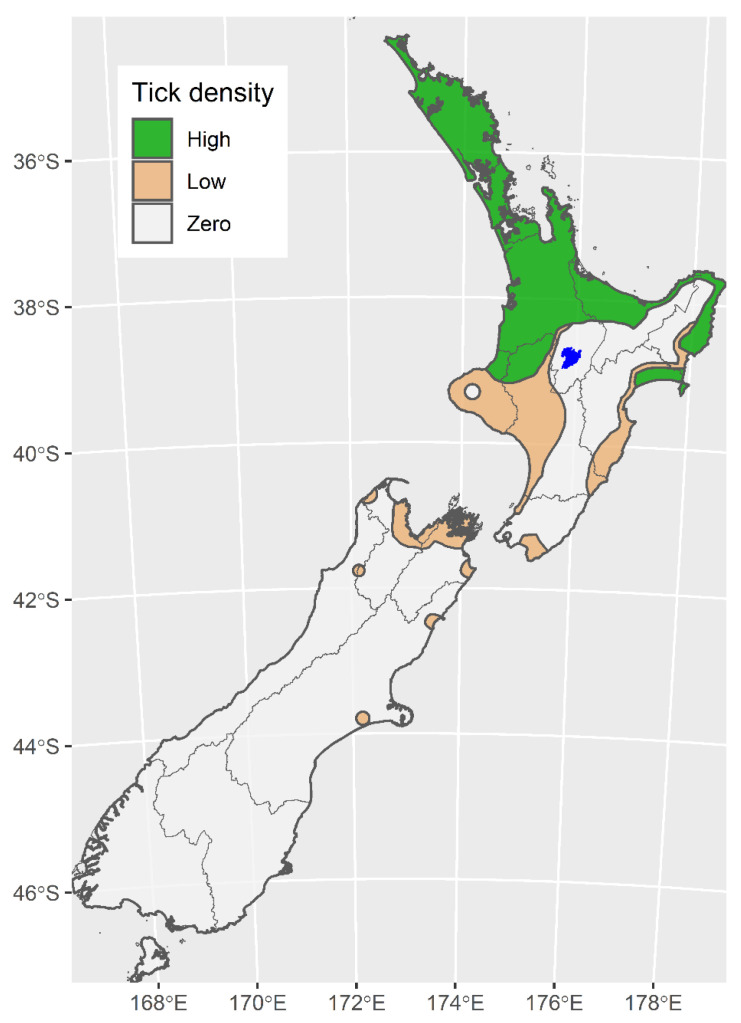
Distribution and density of ticks in New Zealand based on telephone survey, field observation and tick surveillance on deer [9] (Reproduced with permission of Taylor & Francis and Copyright Clearance Center).

**Figure 5 pathogens-10-01346-f005:**
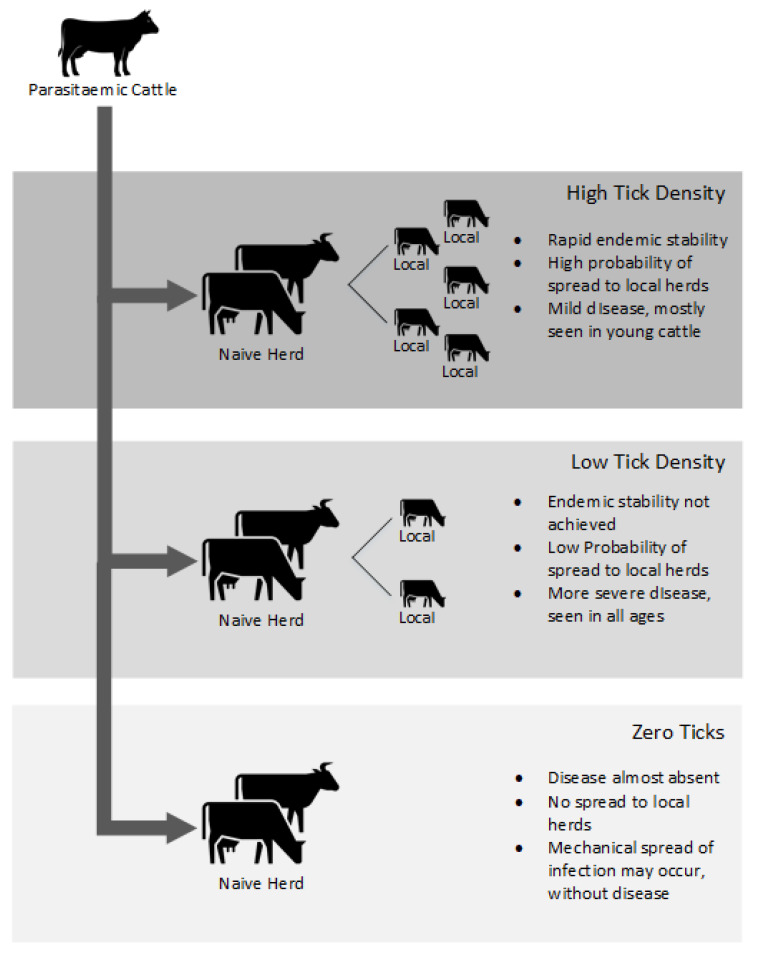
Mechanisms and consequences of *Theileria orientalis* (Ikeda) spread by parasitaemic cattle.

**Figure 6 pathogens-10-01346-f006:**
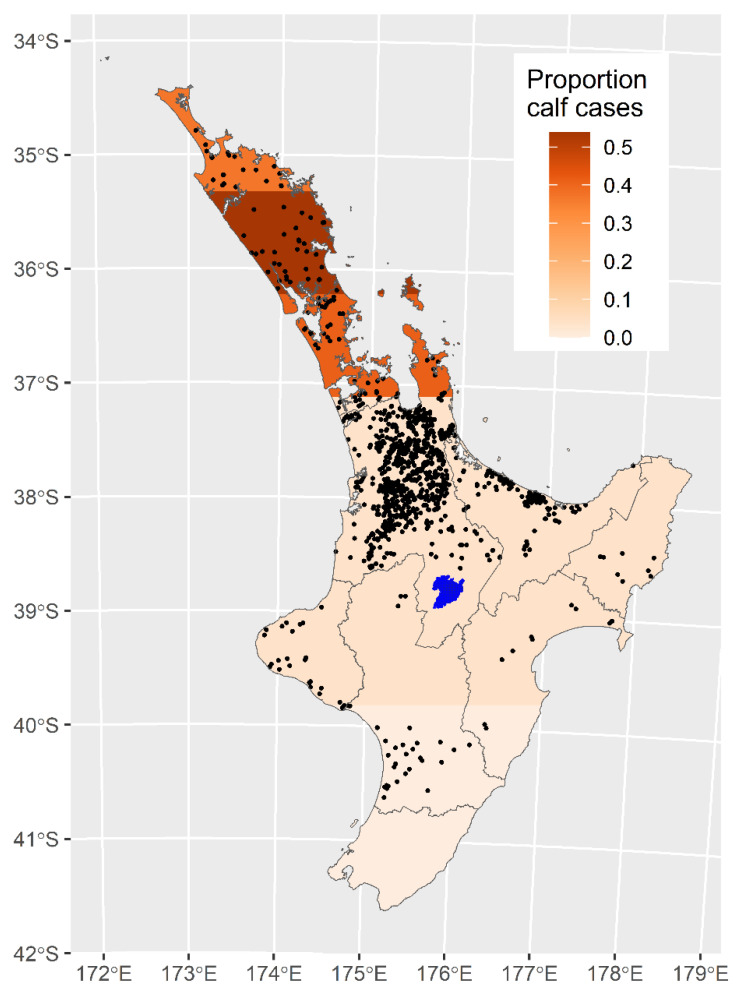
Prevalence of TABA cases in calves by latitude across the North Island, New Zealand. The black dots show the locations of all *Theileria orientalis* Ikeda cases from 22 August 2012 to 1 December 2014.

**Figure 7 pathogens-10-01346-f007:**
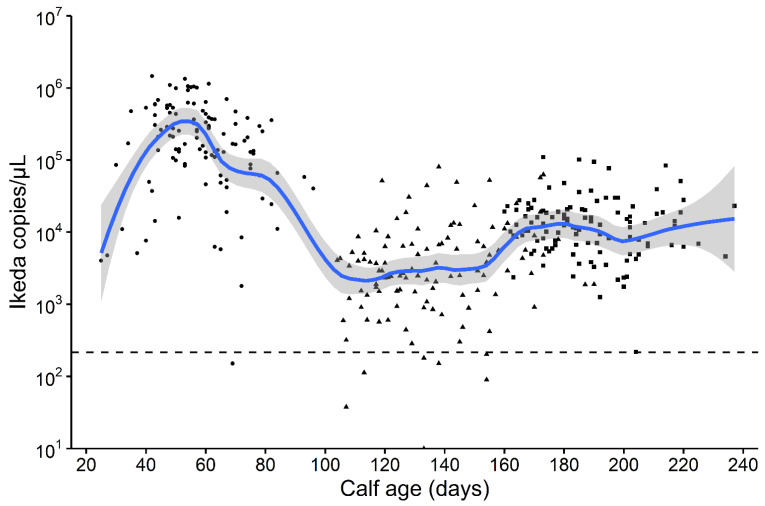
Change in log10 *Theileira orientalis* Ikeda infection intensity per µl blood with age, for Angus calves born in an endemic stable region of New Zealand. Calves were blood sampled in the spring (•), summer (^▲^) and autumn (▪), the broken horizontal line is the limits of detection for this molecular PCR method [20]. A Loess line with 95% confidence interval is fitted to the data using a span = 0.25.

**Figure 8 pathogens-10-01346-f008:**
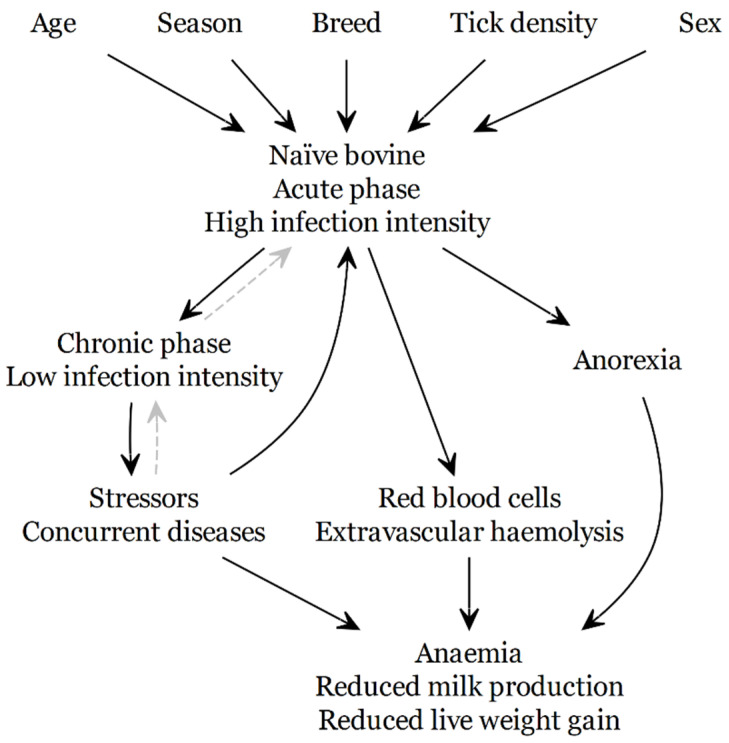
Causal diagram for the pathogenesis of disease, anaemia, live weight, and milk production effects in naïve cattle recently infected with *Theileria orientalis* (Ikeda). The broken grey lines and arrows are used to show a weak association.

**Figure 9 pathogens-10-01346-f009:**
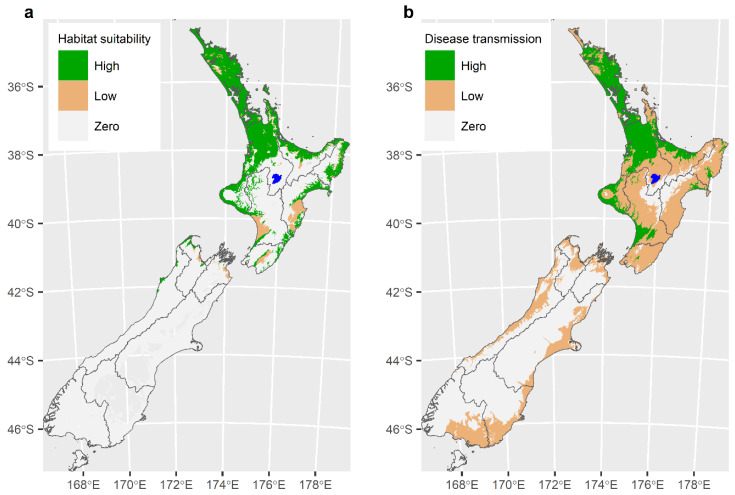
(**a**) The habitat suitability for *Haemaphysalis longicornis* [11] (Reproduced with permission of Elsevier and Copyright Clearance Center). (**b**) The environmental suitability for disease transmission of *Theileria* associated bovine anaemia [13]. (Reproduced with permission of Elsevier and Copyright Clearance Center).

**Figure 10 pathogens-10-01346-f010:**
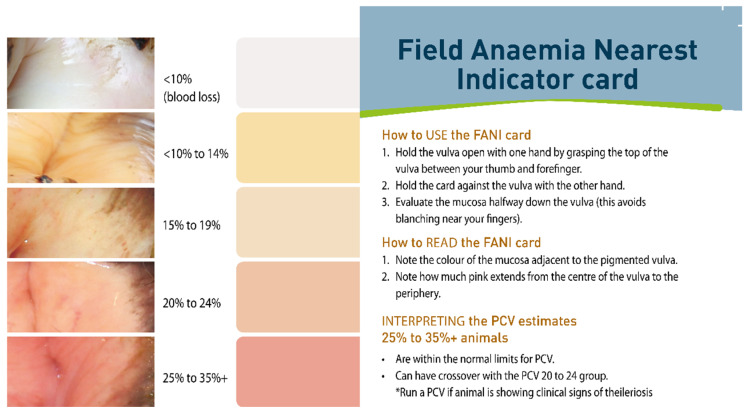
Front side of Field Anaemia Nearest Indicator (FANI) card [source Ministry for Primary Industries, New Zealand].

**Figure 11 pathogens-10-01346-f011:**
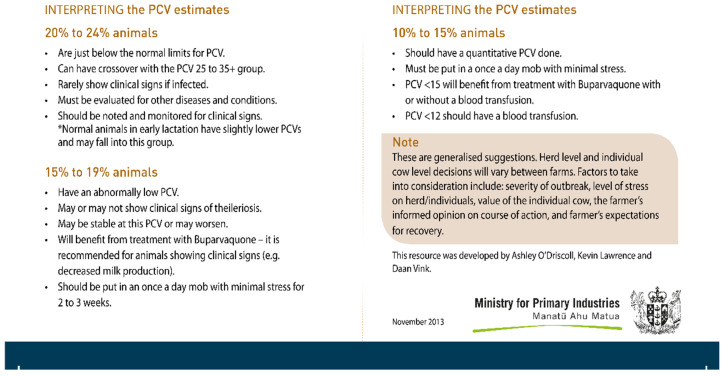
Reverse side of Field Anaemia Nearest Indicator (FANI) card [source Ministry for Primary Industries, New Zealand].

## Data Availability

Not applicable.

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
