# Peer review of "Review of the New Zealand Theileria orientalis Ikeda Type Epidemic and Epidemiological Research since 2012"

_pathogens, 2021, doi:10.3390/pathogens10101346_

Round 1
Reviewer 1 Report
This is interesting and helpful review. It is obvious from Reference List that the authors mainly cited their own observations and researches. However, this is not illegal self-citation. It's just that the authors are the most experienced specialists in this topic.
The available data and ideas are clearly and presented.
However, I have a few suggestions to the authors. I realize that the authors have their own concept of presentation and do not request revisions. Nevertheless, if other reviewers will suggest minor or even major revisions, several improvements would be done. Firstly, it would be nice to "frame" the text presented with an introduction describing the study of this infection before the epidemic in New Zealand, and a conclusion describing what lessons the rest of the world should learn and what threatens due to the existence of endemic areas to infection with T. orientalis Ikeda in New Zealand. Thus, the review would be placed in an international context and would not reflect exclusively New Zealand issues.
Secondly, the authors take a rather pessimistic view of the problem: they believe that the T. orientalis Ikeda infection has come to New Zealand forever and, despite the losses caused, do not set the task of its eradication. Indeed, eradication would be extremely difficult in a large country with long land borders. But in small, highly cultured, high-tech, insular New Zealand - why not dream?
Reviewer 2 Report
A lovely study and review by Kevin Lawrence et al for summarises the New Zealand epidemic and the epidemiological research conducted on the epizootic of bovine anemia associated with Theileria orientalis Ikeda type infection, which began in New Zealand in August 2012. The authors also explain the future study need to address An effective chemotherapeutic agent to reduce animal suffering and lost production for Theileria infection. Overall the manuscript is well written and presents important findings for the prevention and control of livestock infections. I don't have any specific comments or suggestions.
Author Response
Thank you for your kind comments
Reviewer 3 Report
Reviewer’s comments to pathogens-1376567
This review article summarized the bovine oriental theileriosis in New Zealand. I think this theileriosis epidemic and current endemic situation is very important information to understand the emergence and spread of the disease. The paper is well written and most of the figures are clearly presented. I recommend the manuscript for acceptance after minor revision.
- When I finish reading the manuscript I understand the situation, however I was wondering what is the current situation of oriental theileriosis in New Zealand. Since the authors used the data around 2014-2014 in the manuscript, it is sometime hard to imagine the current situation. I recommend to add some explanation.
- Figure8: I think the authors can improve the connection of the key words and presenting style of the key words. ① What “RBCs” means? Regarding anemia, hematopoiesis or other meaning? ② I guess “reduction of” Milk production. If so, it is better to add arrow (↓). ③ I feel (risk) factor, outcome, symptom and infection status are mixing in the diagram. It may better to differentiate the key word category and revise the diagram.
- Page 11, Managing the NZ epidemic: How local veterinarians diagnose the oriental theileriosis? Are they checking the Giemsa stained blood smear or just checking the clinical symptoms? The authors should add some information about diagnosis.
- Page 13, Line 333: Since I’m not a native English speaker, I couldn’t follow the word “post gypsy day”. It may better to change (when I searched, I found one article which said Otago Regional Council won't use the term) or add brief explanation.
Author Response
- When I finish reading the manuscript I understand the situation, however I was wondering what is the current situation of oriental theileriosis in New Zealand. Since the authors used the data around 2014-2014 in the manuscript, it is sometime hard to imagine the current situation. I recommend to add some explanation.
At present in 2021 T orientalis Ikeda remains endemic in all these original areas with veterinarians still reporting clinical cases but not the mass numbers seen with the original epidemic
- Figure8: I think the authors can improve the connection of the key words and presenting style of the key words. ① What “RBCs” means? Regarding anemia, hematopoiesis or other meaning? ② I guess “reduction of” Milk production. If so, it is better to add arrow (↓). ③ I feel (risk) factor, outcome, symptom and infection status are mixing in the diagram. It may better to differentiate the key word category and revise the diagram.
Have revised the diagram a little using some of your suggestions
- Page 11, Managing the NZ epidemic: How local veterinarians diagnose the oriental theileriosis? Are they checking the Giemsa stained blood smear or just checking the clinical symptoms? The authors should add some information about diagnosis.
Have added a sentence to explain what practitioners did
- Page 13, Line 333: Since I’m not a native English speaker, I couldn’t follow the word “post gypsy day”. It may better to change (when I searched, I found one article which said Otago Regional Council won't use the term) or add brief explanation
Have added some explanation and used moving day instead of gypsy day which is a more correct expression